# Acidic Urine pH and Clinical Outcome of Lower Urinary Tract Infection in Kidney Transplant Recipients Treated with Ciprofloxacin and Fosfomycin

**DOI:** 10.3390/antibiotics13020116

**Published:** 2024-01-24

**Authors:** Soraya Herrera-Espejo, Sara Fontserè, Carmen Infante, Alejandro Suárez-Benjumea, Marta Carretero-Ledesma, Marta Suñer-Poblet, Carmen González-Corvillo, Gabriel Bernal, Guillermo Martín-Gutiérrez, Juan Antonio Pérez-Cáceres, Jerónimo Pachón, María Eugenia Pachón-Ibáñez, Elisa Cordero

**Affiliations:** 1Clinical Unit of Infectious Diseases, Microbiology and Parasitology, Institute of Biomedicine of Seville (IBiS), Virgen del Rocio University Hospital/CSIC/University of Seville, 41013 Seville, Spain; sherrera-ibis@us.es (S.H.-E.); guillermo.martin.gutierrez.sspa@juntadeandalucia.es (G.M.-G.); juan.perez.caceres.sspa@juntadeandalucia.es (J.A.P.-C.); mcordero6@us.es (E.C.); 2Institute of Biomedicine of Seville (IBiS), Virgen del Rocio University Hospital/CSIC/University of Seville, 41013 Seville, Spainpachon@us.es (J.P.); 3CIBER de Enfermedades Infecciosas (CIBERINFEC), Instituto de Salud Carlos III, 28029 Madrid, Spain; 4Urology and Nephrology Unit, Virgen del Rocío University Hospital, 41013 Seville, Spain; 5Department of Medicine, School of Medicine, University of Seville, 41004 Seville, Spain

**Keywords:** urinary tract infections, urine pH, ciprofloxacin, fosfomycin, kidney transplant recipients, *Escherichia coli*, *Klebsiella pneumoniae*

## Abstract

Different factors, including antimicrobial resistance, may diminish the effectiveness of antibiotic therapy, challenging the management of post-transplant urinary tract infection (UTI). The association of acidic urine pH with microbiological and clinical outcomes was evaluated after fosfomycin or ciprofloxacin therapy in 184 kidney transplant recipients (KTRs) with UTI episodes by *Escherichia coli* (N = 115) and *Klebsiella pneumoniae* (N = 69). Initial urine pH, antimicrobial therapy, and clinical and microbiological outcomes, and one- and six-month follow-up were assessed. Fosfomycin was prescribed in 88 (76.5%) *E. coli* and 46 (66.7%) *K. pneumoniae* UTI episodes in the total cohort. When the urine pH ≤ 6, fosfomycin was prescribed in 60 (52.2%) *E. coli* and 29 (42.0%) *K. pneumoniae*. Initial urine pH ≤ 6 in *E. coli* UTI was associated with symptomatic episodes (8/60 vs. 0/55, *p* = 0.04) at one-month follow-up, with a similar trend in those patients receiving fosfomycin (7/47 vs. 0/41, *p* = 0.09). Acidic urine pH was not associated with microbiological or clinical cure in *K. pneumoniae* UTI. At pH 5, the ciprofloxacin MIC_90_ increased from 8 to >8 mg/L in *E. coli* and from 4 to >8 mg/L in *K. pneumoniae*. At pH 5, the fosfomycin MIC_90_ decreased from 8 to 4 mg/L in *E. coli* and from 512 to 128 mg/L in *K. pneumoniae*. Acidic urine is not associated with the microbiological efficacy of fosfomycin and ciprofloxacin in KTRs with UTI, but it is associated with symptomatic UTI episodes at one-month follow-up in *E. coli* episodes.

## 1. Introduction

Urinary tract infections (UTIs) remain a major issue in kidney transplant recipients (KTRs), with an incidence of 7.3% and increased mortality in the first year [1,2]. The incidence of UTIs is even higher in the six months (36.5–47%) after transplantation, being the most frequent etiologies *Escherichia coli*, *Klebsiella* spp., and *Pseudomonas aeruginosa* [1,3,4]. In addition to its clinical impact, the treatment of UTI in KTRs poses a serious problem owing to the antimicrobial resistance of the most frequent Gram-negative bacilli [1,3]. In a multicenter European study, the resistance rates among 775 *E. coli* isolates were 15.1% and 1.3% for ciprofloxacin and fosfomycin, respectively, with multidrug resistance (MDR) in 13.9% [5]. In two multicenter studies from China and South Korea, the fosfomycin resistance rates among *E. coli* urine isolates were 3.9% and 5% [6,7]. For *K. pneumoniae*, fosfomycin resistance rates in urine isolates in two Spanish cohorts of KTRs were 22%–25% [8].

A retrospective, multicenter, cohort study evaluated the efficacy of oral fosfomycin in KTRs with cystitis, caused mainly by *E. coli* and *K. pneumoniae* [9]; the clinical and microbiological cure was achieved in 83.9% and 70.2% of cases, respectively. In three clinical trials performed in KTRs, the screening and treatment with different antimicrobials for asymptomatic bacteriuria (AB) caused mostly by *E. coli* and *Klebsiella* spp. did not reduce the occurrence of later symptomatic UTI [10] or acute pyelonephritis [11,12] compared with no treatment and seemed to promote the emergence of resistant organisms [10,12]. In this context, the 2023 EAU Guidelines consider fosfomycin trometamol among the first-line treatments for uncomplicated cystitis and recommend against the screening and treatment of asymptomatic bacteriuria, including in KTRs [13]. Moreover, the EMA recommends avoiding the use of quinolones for mild/moderate bacterial infections and considering patients with organ transplantation at a higher risk [14].

Low-level resistance to ciprofloxacin (LLQR) has been described in *E. coli*, with a frequency of 17% to 39% [15,16]. In LLQR *E. coli* strains, either isogenic derivatives of ATCC 25922, carrying combinations of the most prevalent chromosomal mutations, or uropathogenic LLQR *E. coli* clinical isolates, the acidic pH of the medium increases the ciprofloxacin minimum inhibitory concentration (MIC) against *E. coli* between 32-fold and 256-fold at pH 5 [17]. The same effect has been reported in *E. oli* ATCC 25922 derivatives harboring the *qnrA1*, *qnrB1*, *qnrC*, *qnrD1*, and *qnrS1* genes and in nine *E. coli* clinical strains harboring well-characterized Qnr determinants, with an increase in MIC from 16- to 128-fold when urine pH decreased from 7 to 5 [18]. Other studies using *E. coli* 25922 and *Klebsiella oxytoca* showed that the acidification of urine led to a major impairment of the antimicrobial activity of all tested fluoroquinolones, with near-total neutralization of activity in time-kill experiments [19].

In contrast to ciprofloxacin, the acidic pH of the culture medium seems to increase the activity of fosfomycin [20,21]. Using *E. coli* BW25113 and 10 isogenic strains carrying fosfomycin chromosomal mutations and five fosfomycin-resistant *E. coli* urine isolates, the MIC decreased steadily from pH 7 to pH 5 in most cases; at pH 8, a 2-fold to 16-fold increase in the MIC was observed [20]. Another study showed that the fosfomycin MIC_90_ against 158 *E. coli* urine isolates was lower at pH 6 compared with pH 7, although the pH conditions did not affect the activity against *Klebsiella* spp. [21]. 

Nevertheless, the impact of physiological urine pH changes on the clinical and microbiological efficacy of the therapy in *E. coli* and *K. pneumoniae* lower UTIs has not been analyzed in patients, including KTRs. Thus, in this study, we aimed to evaluate the microbiological and clinical outcomes, during a six-month follow-up period, of the treatment of *E. coli* and *K. pneumoniae* UTIs with fosfomycin and ciprofloxacin in KTRs and determine associations with the urine pH, and also to assess the changes produced at different pH in the MIC distribution of fosfomycin and ciprofloxacin in the urine isolates of *E. coli* and *K. pneumoniae* and in their bactericidal activity.

## 2. Results

### 2.1. Characteristics of KTRs with E. coli and K. pneumoniae UTI Episodes

We included *E. coli* and *K. pneumoniae* UTI episodes in 115 and 69 KTRs, respectively. Among the *E. coli* episodes, 19 (16.5%) and 96 (83.5%) were cystitis and AB, respectively, and among the *K. pneumoniae* episodes, 33 (47.8%) and 36 (52.2%) were cystitis and AB. The most common immunosuppressive drug combination in the two etiologies was mycophenolate (MMF), prednisone, and tacrolimus. Demographics, Charlson comorbidity index, characteristics of the transplantation, data on underlying end-stage renal disease, previous allograft rejection, plasma creatinine and urine pH at inclusion, antimicrobial susceptibilities of *E. coli* and *K. pneumoniae* isolates from the episodes, and antimicrobial therapy are detailed in Table 1. No differences regarding cystitis or AB episodes were found for any variables (Appendix A).

### 2.2. Association of Urine pH with Microbiological and Clinical Outcomes of E. coli and K. pneumoniae UTI Episodes

The 115 patients with *E. coli* UTI episodes were treated with fosfomycin and ciprofloxacin in 88 (76.5%) and 27 (23.5%) cases, respectively. At inclusion, urine pH was acidic (≤6) in 60 (52.1%) episodes. Acidic urine, compared to neutral or alkaline urine, was not associated with microbiological cure one month after therapy (61.7% vs. 69.1%) in all episodes, nor in episodes treated with fosfomycin or ciprofloxacin (Table 2). Regarding clinical outcome, acidic urine at inclusion was associated with more symptomatic events at one-month follow-up (13.3% vs. 0%, *p* = 0.045) in all episodes and in those treated with fosfomycin (14.9% vs. 0%, *p* = 0.013). By the six-month follow-up, renal function had worsened in fourteen (12.3%) patients, one (0.8%) patient lost the graft, and one (0.8%) patient died in the fourth month with acute pyelonephritis by *E. coli* and renal failure.

Among the 69 patients with *K. pneumoniae* UTI episodes, 46 (66.7%) and 23 (33.3%) were treated with fosfomycin and ciprofloxacin, respectively. At inclusion, urine pH was acidic (≤6) in 29 (42.0%) episodes. Acidic urine, compared to neutral or alkaline urine, was not associated with microbiological cure at one month in all episodes, nor in episodes treated with fosfomycin or ciprofloxacin, nor were there differences in symptomatic UTI episodes at one- and six-month follow-up (Table 2). By the six-month follow-up, renal function worsened in seven (10.1%) patients, one (1.4%) patient lost the graft, and four (5.8%) patients died in the second month of follow-up, in the context of bloodstream infections by *K. pneumoniae* in two patients. 

As a sensitivity analysis, in the 184 *E. coli* and *K. pneumoniae* UTI episodes, the urine pH was acidic (≤6) at inclusion in 89 (48.4%) cases, which was associated with more symptomatic events by the one-month follow-up (13.5% vs. 4.2%, *p* = 0.052) (Appendix A).

### 2.3. Antimicrobial and Bactericidal Activities of Ciprofloxacin and Fosfomycin against E. coli and K. pneumoniae Clinical Isolates at Neutral, Acidic, and Alkaline pH

Among the 115 *E. coli* initial isolates, 32 (27.8%) were resistant to ciprofloxacin, and 10 (8.7%) were LLQR. When the MIC of ciprofloxacin was determined at pH 5, the resistance and LLQR rates increased to 40.9%; minimal changes in the MIC occurred at pH 8. Nine (7.8%) isolates were resistant to fosfomycin and three (2.6%) were LLFR. In the MIC determination of fosfomycin at pH 5, there were no appreciable changes; at pH 8, the resistance increased to 15.7%, and 12.2% had LLQR. Regarding the 69 initial *K. pneumoniae* isolates, 23 (33.3%) were resistant to ciprofloxacin, and 13 (18.8%) were LLQR. The MIC determination of ciprofloxacin at pH 5 increased resistance to 49.3%, and 47.8% exhibited LLQR; at pH 8, the MIC_50_ and MIC_90_ values decreased from 0.25 and 4 to 0.03 and 2 mg/L, respectively. Fifty-five (79.7%) isolates were resistant to fosfomycin. When the MIC of fosfomycin was determined at pH 5 and pH 8, the resistance increased to 97.1% and 98.6%, respectively. Details on the MIC distributions of ciprofloxacin and fosfomycin at different pH are in Figure 1 and Appendix A. 

The growth of the *E. coli* and *K. pneumoniae* strains in Müller-Hinton Broth (MHB) or urine was not different depending on the pH, but it was approximately 1 log_10_ CFU/mL lower in urine. Against the *E. coli* strains at neutral pH, ciprofloxacin was bactericidal at 24 h against three and four strains in MHB and urine, respectively; fosfomycin was bactericidal at six hours, with regrowth at 24 h, against three strains in MHB. At acidic pH, ciprofloxacin at 24 h and fosfomycin at 6 h, with regrowth at 24 h, were bactericidal only against one strain, respectively. Finally, at alkaline pH, ciprofloxacin was bactericidal at 24 h against the four strains in MHB and urine, and fosfomycin at six hours, with regrowth at 24 h, against two strains in MHB (Figure 2, Appendix A). In the experiments with regrowth at 24 h, *E. coli* Nu14, at neutral pH, developed resistant mutants to fosfomycin in 60% and 20% of strains, respectively, in MHB and urine, and at alkaline pH, it also showed a 20% mutant resistance in MHB. 

Regarding *K. pneumoniae* strains, ciprofloxacin was bactericidal at 24 h against two strains, at neutral, acidic, and alkaline pH, in both MHB and urine. Fosfomycin, at neutral pH, was bactericidal at 24 h against one strain in both MHB and urine. At acidic pH, fosfomycin was bactericidal at 24 h against one and two strains in MHB and urine, respectively, and at 2 h against one strain in urine, with regrowth at 24 h, without developing resistance to fosfomycin. At alkaline pH, fosfomycin was bactericidal at 24 h against one strain in MHB (Figure 3, Appendix A).

## 3. Discussion

The present study shows that in KTRs with UTI caused by *E. coli*, cystitis, and asymptomatic bacteriuria, the acidic urine pH (≤6) at diagnosis, although physiological, is associated with a higher frequency of symptomatic UTI at one month of follow-up, particularly in those treated with oral fosfomycin. The acidic urine was not associated with the microbiological and clinical cure at one and six months after therapy, respectively, in patients with *E. coli* UTI, nor with the microbiological and clinical outcomes in KTRs with UTI caused by *K. pneumoniae*. At pH 5, the ciprofloxacin MIC_90_ increased in *E. coli* and *K. pneumoniae*. Fosfomycin MIC_90_ increased at pH 8 in *E. coli* and decreased at pH 5 in *K. pneumoniae*. The bactericidal in vitro activity of ciprofloxacin and fosfomycin against the *E. coli* strains decreased at acidic pH. At acidic pH, there was a slight increase of the bactericidal activity in fosfomycin against *K. pneumoniae* without changes with ciprofloxacin.

The clinical outcome of fosfomycin in the present study is inconsistent with its reported pharmacodynamics, including urine acidification. Using a Monte Carlo simulation of the urinary fosfomycin area under the concentration–time curve, after a single oral dose of 3 g, fosfomycin was effective against *E. coli* (MIC_90_ ≤ 16 mg/L) but not against *Klebsiella* spp. Acidification increased the susceptibility of 71% of the bacterial isolates, and the cumulative fractions of the bacterial responses were 99% and 55% against *E. coli* and *Klebsiella* spp., respectively, based on simulated drug exposure in urine with an acidic pH of 6 [21]. However, a retrospective study, including 48 cases of asymptomatic bacteriuria (AB) and cystitis, found fosfomycin resistance after treatment in six (12.5%) episodes caused by *Enterobacterales* [22], as we have observed against one *E. coli* strain after exposition to fosfomycin in the time-kill assays. In addition, a multicenter study showed 9.1% heteroresistance to fosfomycin among 66 *E. coli* urine isolates, with overexpression of metabolic genes increasing their survival rate [23], which may play a role in the failure of antibiotic treatments. 

The clinical outcomes in the present study did not confirm the in vitro studies showing that the acidic pH of the medium increased the ciprofloxacin MIC against *E. coli* in LLQR [17], strains harboring well-characterized Qnr determinants [18], or *E. coli* 25922 [19]. Although our in vitro studies showed an increase in the MIC90 and a decrease in the susceptibility rate to ciprofloxacin in *E. coli* at diagnosis, the clinical and microbiological outcomes in patients treated with ciprofloxacin did not differ depending on the pH (acidic vs. neutral or alkaline).

The time-kill results were in accordance with those previously published [17,18,19], showing that the growth of *E. coli* and K. *pneumoniae* strains was similar, independent of pH conditions, and that *E. coli* growth was lower in urine than in MHB [18,19,20]. As reported [17,18,19], the present study shows less bactericidal in vitro activity of ciprofloxacin against *E. coli* in acidic pH conditions, independent of the medium. Likewise, at acidic pH conditions, fosfomycin activity against *E. coli* and *K. pneumoniae* was found to be marginal. Burian et al. [24] also found that pH acidification decreases the activity of different antibiotics, including fosfomycin. Martín-Gutiérrez et al. [20] found reduced fosfomycin activity in MHB at alkaline pH with an increase in the fosfomycin MIC against susceptible *E. coli* strains and those with LLFR. Nevertheless, in our in vitro studies, fosfomycin was bactericidal against three and two out of four isolates at neutral and alkaline pH, respectively, and at acidic pH only against one isolate. 

Our study had several limitations. It was not a controlled study, although it reflected the results obtained in daily clinical practice. The number of patients treated with ciprofloxacin was limited following the EMA recommendation [14]. Moreover, the small number of patients with symptomatic UTIs in the follow-up precluded the exploration of possible confounding variables. The strengths were the inclusion of 184 UTI episodes by *E. coli* and *K. pneumoniae* and treatment only with fosfomycin or ciprofloxacin, considering that clinical trials to evaluate the therapeutic efficacy in AB included 112 and 205 KTRs with UTI of any etiology and receiving nonhomogeneous therapies [10,11,12]. Moreover, physiologically acidic urine was a common event, occurring in 48.4% of the 184 KTRs in the present study. 

The study had several implications for clinical practice. First, it pointed out the need to consider an initial acidic urine pH as a factor for the strict follow-up of KTRs with *E. coli* UTI, especially in those treated with fosfomycin. Regarding the future, controlled and randomized clinical trials may answer the question of what the better therapy for cystitis is in KTRs with the most frequent gram-negative bacilli etiologies. 

## 4. Materials and Methods

### 4.1. Study Design and Setting

We carried out an observational cohort of adult KTRs with *E. coli* and *K. pneumoniae* UTI episodes (cystitis and asymptomatic bacteriuria ([AB]), who attended as outpatients at the Virgen del Rocío University Hospital, Seville, Spain, from January 2017 to December 2019 and in a second period from March 2021 to June 2022 to collect additional *K. pneumoniae* episodes.

Physicians in charge of patients asked them to participate in the study, and they were prospectively included if (i) they provided informed consent; (ii) the requested urine cultures identified *E. coli* or *K. pneumoniae*; (iii) the attendant physicians prescribed fosfomycin or ciprofloxacin therapy in accordance with their clinical criteria; (iv) urine cultures were performed between 14 and 30 days after beginning the treatment; and (iv) a six-month follow-up was available, with an optional urine culture if new UTI symptoms occurred. Patients who did not fulfill these criteria were excluded. Fosfomycin trometamol was administered as two oral doses of 3 g administered 48 h apart [25], and ciprofloxacin was administered as 250 mg orally every 12 h for 5 days [26]. The outcomes investigated were microbiological cure at one month and clinical cure at one- and six-month follow-ups.

The following data were recorded from the digital charts at inclusion: demographics, chronic underlying diseases, time since kidney transplantation, immunosuppressive regimens, clinical data, plasma creatinine, urinary pH (pH ≤ 6 was defined as acidic), leukocyturia, urine nitrites in samples processed 4–8 h after collection, and antimicrobial therapy. The GESITRA/REIPI UTI guidelines were followed for clinical definitions [3]. Bacteriuria was defined as urine specimens isolated with quantitative counts of ≥10^5^ CFU/mL. Asymptomatic bacteriuria was defined as the presence of bacteriuria in the absence of any UTI symptoms. Cystitis was considered for bacteriuria and clinical manifestations such as dysuria, frequency and urgency of urination, suprapubic pain, and/or hematuria in the absence of pyelonephritis symptoms. Acute pyelonephritis (APN) was considered the simultaneous presence of bacteriuria and/or bacteremia and fever, with one or more of the following: lumbar pain (if native kidney involved), renal allograft tenderness (if the kidney was transplanted), chills, or cystitis symptoms. Microbiological cure was achieved when urine culture was negative at 14–30 days. Clinical cure was considered the resolution of symptoms in the case of cystitis. Mortality was considered as death occurring within six months of follow-up. Impairment of renal function was defined as a ≥0.5 mg/dL increase in plasma creatinine.

### 4.2. Antimicrobial Susceptibility of E. coli and K. pneumoniae Clinical Isolates at Different pH Conditions

Clinical isolates were collected at inclusion and during follow-up and processed within 4–8 h after collection, and urine pH was measured. The hospital microbiology service identified the bacterial isolates and performed susceptibility testing with standard tests. Causative organisms were identified using a MicroScan WalkAway^®^ Plus system (Beckman Coulter, Nyon, Switzerland). In cases where identification was uncertain, verification was obtained using a Bruker Biotyper MALDI-TOF MS system (Bruker Daltonik GmbH, Leipzig, Germany). The antimicrobial susceptibility testing and interpretation were in accordance with the yearly European Committee on Antimicrobial Susceptibility Testing (EUCAST) criteria [27].

Moreover, ciprofloxacin and fosfomycin MICs were determined in duplicate, in solutions with pH 8, 7, and 5 and interpreted following the 2023 EUCAST criteria (www.eucast.org/clinical_breakpoints, accessed on 22 December 2023): in the absence of defined breakpoints for oral fosfomycin in *Enterobacterales* other than *E. coli*, the same criteria were applied for *K. pneumoniae*. Antimicrobials were purchased as standard powders (Sigma-Aldrich, Madrid, Spain). The in vitro ciprofloxacin susceptibility assay was determined by microdilution method adjusting MHB (Thermo Scientific, Oxford, UK) to obtain acidic (pH = 5) or alkaline (pH = 8) pH values by adding 0.012% (*v*/*v*) of 12 N HCl or 0.072% (*v*/*v*) of 2 N NaOH, respectively (Sigma-Aldrich, Spain). Increasing concentrations of ciprofloxacin (from 0.01 to 8 mg/L) were tested with a starting inoculum of 5 × 10^5^ CFU/mL. Fosfomycin susceptibility assays were carried out by agar diffusion. Aliquots of LB-agar (Sigma-Aldrich, Spain) were supplemented with glucose-6-phosphate (25 mg/L), increasing concentrations of fosfomycin (from 0.12 to 1024 mg/L) and, finally, 0.05% (*v*/*v*) of 12 N HCl or 0.07% (*v*/*v*) of 10 N NaOH to adjust acidic (pH = 5) or alkaline (pH = 8) pH values, respectively. Aliquots were plated and gel, then 2 µL of inoculum (final concentration of 5 × 10^5^ CFU/mL) was plated and incubated overnight at 37 °C. For *E. coli*, ciprofloxacin, and fosfomycin with MIC values of >0.06 to 0.5 mg/L and >4 to 8 mg/L were considered the LLQR and LLFR, respectively. For *K. pneumoniae*, ciprofloxacin MIC values of >0.12 to 0.5 mg/L were considered the LLQR; no isolates were classified as LLFR because of the 128 mg/L ECOFF (https://mic.eucast.org/; accessed on 21 July 2023).

### 4.3. Bactericidal Activity of Ciprofloxacin and Fosfomycin against E. coli and K. pneumoniae Strains at Different pH Conditions

To evaluate the impact of different pH conditions on the bactericidal activities of ciprofloxacin and fosfomycin against uropathogenic *E. coli* and *K. pneumoniae* strains, with different susceptibility patterns, we used eight strains: (i) *E. coli* Nu14 [28], ciprofloxacin- and fosfomycin-susceptible (MIC 0.03 and 2 mg/L, respectively); (ii) *E. coli* HUVR94, ciprofloxacin- and fosfomycin-susceptible (MIC 0.03 and 0.5 mg/L, respectively); (iii) *E. coli* Nu79 *gyrA* D87G with LLQR (MIC 0.12 mg/L) and fosfomycin-susceptible (MIC 0.5 mg/L) [29]; (iv) *E. coli* Nu14 with a *glpT* missense mutation, ciprofloxacin-susceptible (MIC 0.01 mg/L) and previously defined as LLFR (MIC 32 mg/L) [30]; (v) *K. pneumoniae* HUVR42, ciprofloxacin- and fosfomycin-susceptible (MIC 0.007 and 4 mg/L, respectively); (vi) *K. pneumoniae* HUVR5, ciprofloxacin- and fosfomycin-resistant (MIC 8 and 128 mg/L, respectively); (vii) *K. pneumoniae* HUVR110, ciprofloxacin-resistant (MIC 8 mg/L) and fosfomycin-susceptible (MIC 4 mg/L); and (viii) *K. pneumoniae* HUVR91, ciprofloxacin-susceptible (MIC 0.06 mg/L) and fosfomycin-resistant (MIC 64 mg/L).

The bactericidal activity of both antimicrobials against the eight strains was determined, in triplicate, by time-kill assays at concentrations equivalent to their MIC on MHB (Thermo-Scientific, UK) and urine from three healthy volunteers who had not undergone antibiotic treatment in the previous three months. Urine samples were pooled and sterilized by filtration (polyether-sulphone membrane filters, 0.22 mm, VWR; Leicestershire, UK) and stored at 4 °C until analysis. MHB and/or urine were adjusted to obtain acidic (pH = 5) or alkaline (pH = 8) pH values by adding 0.012% (*v*/*v*) of 12 N HCl or 0.072% (*v*/*v*) of 2 N NaOH, respectively (Sigma-Aldrich, Spain). The initial concentration of inoculum was 5 × 10^5^ CFU/mL, and samples were taken at 0, 2, 4, 6, and 24 h. Bactericidal activity was defined as a decrease of ≥3 log_10_ CFU/mL based on the initial concentration of inoculum [27].

In these assays, mutant-resistant development was determined if the strains were ciprofloxacin- or fosfomycin-susceptible, and there was regrowth at 24 h after a bactericidal effect at previous time points. In these cases, five colonies were randomly picked up to perform susceptibility assays, as previously described. The mutant resistant rate was calculated as (number of resistant colonies/total number of colonies) × 100.

### 4.4. Statistical Analysis

Continuous variables are expressed as median and interquartile range (IQR), and qualitative variables as proportions. The microbiological cure at one month and clinical cure at one- and six-months follow-up were analyzed separately in patients with *E. coli* or *K. pneumoniae* episodes and treated with ciprofloxacin or fosfomycin, comparing the results in patients with urine pH at diagnosis of ≤6 and >6, through chi-square and Fisher exact tests. The association of urine pH with the microbiological and clinical outcomes was adjusted by age (≤60 vs. >60 years) and cystitis or AB. Several sensitivity analyses were performed to compare patients with urine pH ≤ 6 and > 6 in all patients (N = 184), patients with cystitis (N = 52), and patients with AB (N = 132), independently of the etiology. The bactericidal activities of ciprofloxacin and fosfomycin in time-kill studies are presented as differences in the log_10_ CFU/mL with respect to the initial bacterial concentrations. A *p* value of <0.05 was considered significant. The statistical package SPSS v24.0 (SPSS Inc., Chicago, IL, USA) was used.

## 5. Conclusions

The results of the present study suggest that low urine pH is associated with a greater frequency of symptomatic UTI in the first month after the antimicrobial treatment when the etiology of the initial episode is *E. coli*. However, the data in the follow-up show that acidic urine does not affect the microbiological efficacy of ciprofloxacin and fosfomycin in KTRs with cystitis and asymptomatic bacteriuria.

## Figures and Tables

**Figure 1 antibiotics-13-00116-f001:**
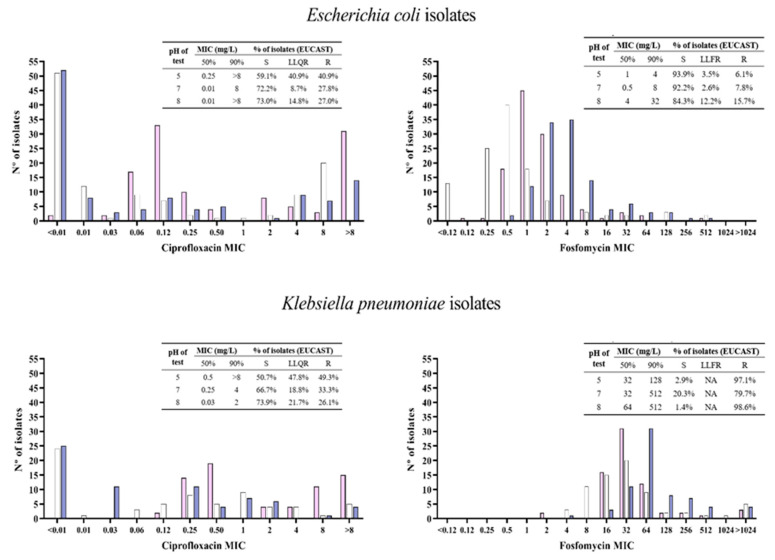
Ciprofloxacin and fosfomycin MIC values distributions determined at pH 5 (pink bars), pH 7 (white bars), and pH 8 (blue bars) against *Escherichia coli* (N = 115) and *Klebsiella pneumoniae* (N = 69) urine clinical isolates. S: susceptible; LLQR: Low-level quinolone resistance; LLFR: Low-level fosfomycin resistance; R: resistant; EUCAST: European Committee on Antimicrobial Susceptibility Testing interpretative criteria 2023 (ciprofloxacin resistant: MIC > 0.5 g/L; fosfomycin resistant: MIC > 8 g/L).

**Figure 2 antibiotics-13-00116-f002:**
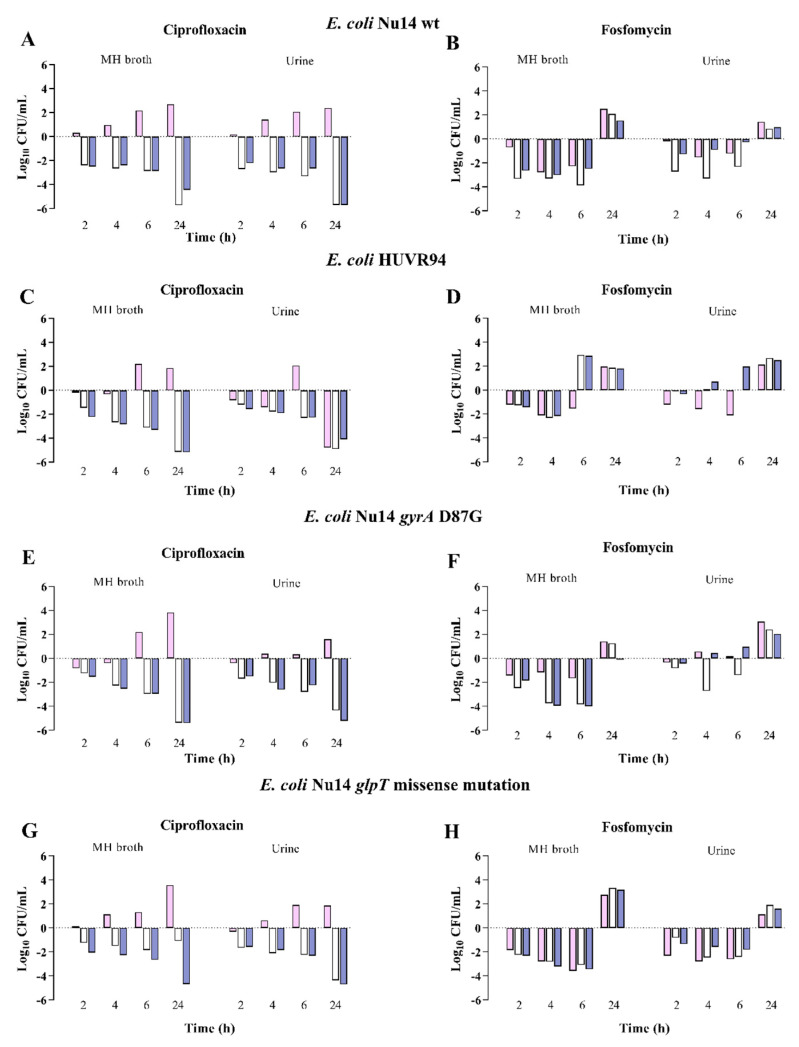
Bactericidal activity of ciprofloxacin and fosfomycin at MIC concentrations in MHB and urine, determined at pH 5 (pink bars), pH 7 (white bars), and pH 8 (blue bars) (dark grey bars) against *Escherichia coli* strains. Panels (**A**,**B**) *E. coli* NU14 wild-type strain, susceptible to ciprofloxacin (MIC 0.03 mg/L) and fosfomycin (MIC 2 mg/L). Panels (**C**,**D**) *E. coli* HUVR94 clinical strain, susceptible to ciprofloxacin (MIC 0.03 mg/L) and fosfomycin (MIC 0.5 mg/L). Panels (**E**,**F**) *E. coli* Nu79 *gyrA* (D87G) strain with low-level quinolone resistance (MIC 0.12 mg/L) and susceptible to fosfomycin (MIC 0.5 mg/L). Panels (**G**,**H**) *E. coli* Nu14 *glpT* missense mutation strains with low-level fosfomycin resistance (MIC 32 mg/L) and susceptible to ciprofloxacin (MIC 0.01 mg/L). Results are represented as differences (log_10_ CFU/mL) relative to the initial time-point (0 h).

**Figure 3 antibiotics-13-00116-f003:**
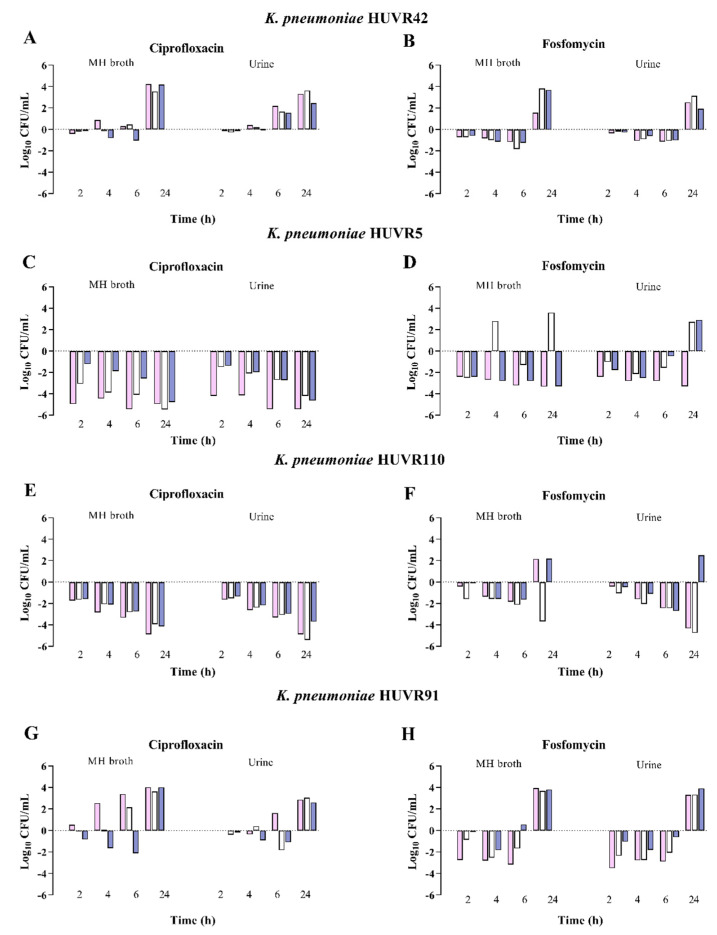
Bactericidal activity of ciprofloxacin and fosfomycin at MIC concentrations in MHB and urine, determined at pH 5 (pink bars), pH 7 (white bars), and pH 8 (blue bars) against *Klebsiella pneumoniae* strains. Panels (**A**,**B**) *K. pneumoniae* HUVR42 susceptible to ciprofloxacin (MIC 0.007 mg/L) and fosfomycin (4 mg/L). Panels (**C**,**D**) *K. pneumoniae* HUVR5 resistant to ciprofloxacin (MIC 8 mg/L) and fosfomycin (128 mg/L. Panels (**E**,**F**) *K. pneumoniae* HUVR110 resistant to ciprofloxacin (MIC 8 mg/L) and susceptible to fosfomycin (MIC 4 mg/L). Panels (**G**,**H**) *K. pneumoniae* HUVR91 resistant fosfomycin (MIC 64 mg/L) and susceptible to ciprofloxacin (MIC 0.06 mg/L). Results are represented as differences (log_10_ CFU/mL) relative to the initial time-point (0 h).

**Table 1 antibiotics-13-00116-t001:** Demographics and characteristics of the kidney transplant recipients with urinary tract infection by *Escherichia coli* and *Klebsiella pneumoniae*.

Variables	*E. coli* 115 Episodes N (%)	*K. pneumoniae* 69 Episodes N (%)
Age (years; median [IQR])	58 (50–67)	61 (50–69)
Female patients	71 (61.7)	42 (60.9)
Charlson Comorbidity Index (median [IQR])	3 (3–5)	5 (3–5)
Months from transplantation (median [IQR])	14 (4–77)	6 (1–77)
<2 months from transplantation	21 (18.3)	41 (59.4)
Previous kidney transplantation	10 (8.7)	8 (11.6)
Living donor	11 (9.5)	6 (8.7)
Induction therapy within 3 previous months: - Thymoglobulin - Basiliximab - Daclizumab	73 (63.5) 26 (22.6) 41 (35.7) 6 (5.2)	28 (40.6) 12 (17.4) 14 (20.3) 2 (2.9)
Current immunosuppression: - Corticosteroids - Tacrolimus - MMF - mTOR inhibitors - Cyclosporine	107 (93.0) 107 (93.0) 90 (78.3) 8 (6.9) 4 (3.4)	58 (84.1) 64 (92.8) 54 (78.3) 4 (5.8) 3 (4.3)
Acute rejection within the previous 6 months	11 (9.6)	0 (0.0)
Rejection treatment in the previous 6 months: - Corticosteroid’s bolus - Plasmapheresis - Thymoglobulin	9 (7.9) 1 (0.9) 1 (0.9)	- - -
Creatinine (mg/dL; median [IQR])	1.57 (1.21–1.95)	1.56 (1.25–1.99)
Bacteriuria within the previous 6 months	57 (49.6)	49 (71.0)
Antibiotic use within the previous 3 months - Quinolones * - Amoxicillin-clavulanate - Fosfomycin - Cephalosporins ** - Others ***	48 (41.7) 11 (9.6) 6 (5.2) 14 (12.2) 15 (13.0) 2 (1.7)	30 (43.5) 3 (4.3) 8 (11.6) 10 (14.5) 7 (10.1) 2 (2.9)
Cystitis	19 (16.5)	33 (47.8)
Asymptomatic bacteriuria	96 (83.5)	36 (52.2)
Urinary pH (median [IQR])	6 (6–6.5)	6.5 (6–6.5)
Baseline antibiotic resistance: - Cotrimoxazole - Ciprofloxacin - Amoxicillin-clavulanate - Fosfomycin - Cephalosporins **** - ESBL-production	66 (57.4) 30 (26.1) 24 (20.9) 10 (8.7) 2 (1.7) 2 (1.7)	42 (60.9) 20 (29.0) 15 (21.7) 8 (11.6) 21 (30.4) 20 (29.0)
Antibiotic therapy of the UTI episodes - Fosfomycin - Ciprofloxacin	88 (76.5) 27 (23.5)	46 (66.7) 23 (33.3)

IQR: Interquartile range; MMF: Mycophenolate mofetil; mTOR inhibitors: Sirolimus, everolimus; ESBL: Extended spectrum beta-lactamases; * Quinolones: ciprofloxacin or levofloxacin; ** Cephalosporins: cefixime or cefuroxime; *** Others: Ertapenem, cloxacillin and rifaximine; **** Cephalosporins: cefuroxime, cefotaxime, ceftazidime, cefixime or cefepime.

**Table 2 antibiotics-13-00116-t002:** Microbiological and clinical outcomes, in patients with acidic vs. non-acidic urine, after fosfomycin or ciprofloxacin therapy of urinary tract infection by *Escherichia coli* and *Klebsiella pneumoniae.*

Variable		Urinary pH ≤ 6	Urinary pH > 6	*p*
		N (%)	N (%)	
***Escherichia coli* UTI Episodes (N = 115)**				
Microbiological cure during one-month follow-up	Total	37/60 (61.7)	38/55 (69.1)	0.41
	Episodes treated with fosfomycin	29/47 (61.7)	27/41 (65.9)	0.69
	Episodes treated with ciprofloxacin	8/13 (61.5)	11/14 (78.6)	0.42
Symptomatic UTI during one-month follow-up	Total	8/60 (13.3)	0/55 (0.0)	0.006
	Episodes treated with fosfomycin	7/47 (14.9)	0/41 (0.0)	0.013
	Episodes treated with ciprofloxacin	1/13 (7.7)	0/14 (0.0)	0.48
Symptomatic UTI during six-month follow-up	Total	11/60 (18.3)	9/55 (16.4)	0.78
	Episodes treated with fosfomycin	10/47 (21.3)	9/41 (22.0)	0.94
	Episodes treated with ciprofloxacin	1/13 (7.7)	0/14 (0.0)	0.48
***Klebsiella pneumoniae* UTI episodes (N = 69)**				
Microbiological cure during one-month follow-up	Total	10/29 (34.5)	15/40 (37.5)	0.69
	Episodes treated with fosfomycin	4/16 (25.0)	8/30 (26.7)	1.00
	Episodes treated with ciprofloxacin	6/13 (46.2)	7/10 (70.0)	0.16
Symptomatic UTI during one-month follow-up	Total	4/29 (13.8)	4/40 (10.0)	0.71
	Episodes treated with fosfomycin	3/16 (18.8)	4/30 (13.3)	0.69
	Episodes treated with ciprofloxacin	1/13 (7.7)	0/10 (0.0)	1.00
Symptomatic UTI during six-month follow-up	Total	3/29 (10.3)	6/40 (15.0)	0.75
	Episodes treated with fosfomycin	1/16 (6.3)	4/30 (13.3)	1.00
	Episodes treated with ciprofloxacin	2/13 (15.4)	2/10 (20.0)	1.00

*p*: Chi-square or Fisher exact tests.

## Data Availability

The raw data supporting the conclusions of this article will be made available by the authors on request.

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
