# Peer review of "Acidic Urine pH and Clinical Outcome of Lower Urinary Tract Infection in Kidney Transplant Recipients Treated with Ciprofloxacin and Fosfomycin"

_antibiotics, 2024, doi:10.3390/antibiotics13020116_

Round 1

Reviewer 1 Report

Comments and Suggestions for Authors

Acidic urine pH and clinical outcome of lower urinary tract infection in kidney transplant recipients treated with ciprofloxacin and fosfomycin.

The significance of antimicrobial resistance (AMR) in kidney transplant recipients cannot be overemphasized since these individuals have a distinct spectrum of difficulties in handling infections. After undergoing a kidney transplant, patients are usually offered immunosuppressive drugs to prevent the rejection of the organ, which increases their vulnerability to infections. The rise of antimicrobial resistance presents a substantial danger by reducing the efficacy of mainstream antibiotic therapies, hence complicating the control of post-transplant infections. Within this susceptible population, it is even more essential to comprehend and tackle the complexities of antimicrobial resistance. This highlights the necessity for continuous research, monitoring, and a multidisciplinary strategy to reduce the consequences of antibiotic resistance in the setting of kidney donation. The manuscript is well constructed to support this hypothesis that the emergence of AMR in kidney transplant recipients having lower urinary tract infections. The following are the few suggestions to improve the manuscript:

Line #

Comment

Abstract

A line may be added in the beginning of the abstract to emphasize the importance of the topic.

Fosfomycin was prescribed in 88 (76.5%) E. coli and 46 (66.7%) K. pneumoniae UTI episodes with urine pH ≤6 in 60 (52.2%) and 29 (42.0%), respectively.

The sentence may be separated in two, for better understanding.

….with a trend in those patients receiving fosfomycin…..

You meant with a similar trend?

…At pH 5, the fosfomycin MIC90 decreased from 512 to 128 mg/L in K. pneumoniae….

At pH 5, what about MIC in E. coli?

36

with an incidence of 7.3% in the first year and higher in the six months after transplantation,

Please rephrase for better understanding …..

44

urine isolates was 3.9 and 5%

urine isolates was 3.9% and 5%

63

E. coli ATCC 25922

E. coli ATCC 25922

87

We included E. coli and K. pneumoniae UTI episodes in 115 and 69 KTRs, respectively.

Was there any inclusion or exclusion criteria defined for the selection of samples?

Mentioned in lines (298-306)

From Table 1:

Antibiotic therapy of the UTI episodes

- Fosfomycin                88 (76.5)          27 (23.5)

- Ciprofloxacin             46 (66.7)          23 (33.3)

We may assume that only patients with prescription of ciprofloxacin and Fosfomycin were selected for this study only.

107-108

…..ciprofloxacin in 89 (77.4%)…..

There seem to be a typo…

You mean 88 (76.5%) as per Table 1???

186

MHB

“Mueller–Hinton broth”

Full form may be written here as it is used here for the first time

Good Luck!

Comments on the Quality of English Language

Some of the sentences can be separated to ease out the readers. Overall, the quality of the English is good.

Author Response

Good afternoon,

Please find attached a document responding to each comment individually.

We hope, that the clarifications/modifications are appropriate.

Kindest regards,

María Eugenia

Reviewer 2 Report

Comments and Suggestions for Authors

The manuscript is well-structured, honestly written about the limitations of the study. The subsection on clinical significance is valuable. The topic of the influence of the pH of the media on the growth of bacteria and the interpretation of the results of resistance to antibiotics is not new. This finding makes a contribution to this topic + to clinical usage of it, although without significant new results.

My recommendation: Instead of Creatinine (mg/dL; median [IQR]), it is worth using the Glomerular filtration rate (GFR) indicator, which is more individual.

Author Response

We fully agree with Reviewer 2 about the significance of the GFR, and we attempted to incorporate this indicator in our analysis. Unfortunately, we were unable to calculate it as the height and, specially, the weight values were not recorded in the clinical chart of the patients on the day of inclusion in the study. This is the reason why we only included the creatinine values.

Reviewer 3 Report

Comments and Suggestions for Authors

1-  This is an important manuscript addressing the effects of UTI  (Escherichia coli, Klebsiella spp, and Pseudomonas aeruginosa)  in Kidney transplant patients, treated with ciprofloxacin and fosfomycin. 

2- At best, this is a very difficult topic, with complications along the way to therapy.

3- However, the authors presented a very credible account by using follow up of UTI at one- and 6-month follow-up to understand the effects of treatment.

4-  The results are well documented in the Figures 2 and 3 mainly. 

Author Response

We appreciate your comments on our work.